# No evidence for the use of magnetic declination for migratory navigation in two songbird species

Nikita Chernetsov[1,2]*, Alexander Pakhomov[1], Alexander Davydov[1], Fedor Cellarius[3], Henrik Mouritsen[4,5]

**1** Biological Station Rybachy, Zoological Institute RAS, St. Petersburg, Russia, **2** Dept. Vertebrate Zoology, St. Petersburg State University, St. Petersburg, Russia, **3** Dept. Vertebrate Zoology, Lomonosov Moscow State University, Moscow, Russia, **4** AG Neurosensory Sciences/Animal Navigation, Institut für Biologie und Umweltwissenschaften, Carl-von-Ossietzky Universität Oldenburg, Oldenburg, Germany, **5** Research Center for Neurosensory Sciences, University of Oldenburg, Oldenburg, Germany

* nikita.chernetsov@gmail.com

**Data Availability Statement:** All relevant data are within the paper and its Supporting Information files.

## Abstract

Determining the East-West position was a classical problem in human sea navigation until accurate clocks were manufactured and sailors were able to measure the difference between local time and a fixed reference to determine longitude. Experienced night-migratory songbirds can correct for East-West physical and virtual magnetic displacements to unknown locations. Migratory birds do not appear to possess a time-different clock sense; therefore, they must solve the longitude problem in a different way. We showed earlier that experienced adult (but not juvenile) Eurasian reed warblers (*Acrocephalus scirpaceus*) can use magnetic declination (the difference in direction between geographic and magnetic North) to solve this problem when they were virtually displaced from Rybachy on the eastern Baltic coast to Scotland. In this study, we aimed to test how general this effect was. Adult and juvenile European robins (*Erithacus rubecula*) and adult garden warblers (*Sylvia borin*) under the same experimental conditions did not respond to this virtual magnetic displacement, suggesting significant variation in how navigational maps are organised in different songbird migrants.

## Introduction

Many millions of avian migrants show fidelity to their breeding and non-breeding areas year after year. To find their preferred locations after spending several months travelling hundreds or thousands of kilometres, they need the ability to find their goal from beyond the range of direct sensory contact, i.e. they must be able to use true navigation [1–3]. True navigation across the surface of the Earth requires the ability to detect at least two non-parallel coordinates, broadly equivalent to what humans call geographic latitude and longitude [1–3]. Measuring these two coordinates is not equally difficult [1, 4]. Even though geographic latitude can be inferred e.g. from the elevation of the centre of stellar rotation (e.g. the North Star) on any

**Funding:** N.C. Grant number 17-14-01147 Russian Science Foundation; https://rscf.ru/en/ N.C. Grant number 18-04-00265 Russian Foundation for Basic Research; https://www.rfbr.ru/rffi/eng A.P. Grant number 20-04-01059 Russian Foundation for Basic Research; https://www.rfbr.ru/rffi/eng H. M. Grant number SFB 1372 Deutsche Forschungsgemeinschaft; https://www.dfg.de/en/index.jsp The funders had no role in study design, data collection and analysis, decision to publish, or preparation of the manuscript.

**Competing interests:** The authors have declared that no competing interests exist.

date, the elevation of the sun at solar noon (when considering season), or from magnetic inclination, geographic longitude is less straightforward to measure. Human long-distance navigators did not know how to determine geographic longitude accurately until the 18th century, when very accurate sea-clocks were manufactured, making it possible to determine the position along the east-west axis from the difference between local time and fixed Greenwich time [5].

Migratory birds are, however, capable of detecting their position along the east-west axis [6, 7]. We have earlier shown that Eurasian reed warblers (*Acrocephalus scirpaceus*) can detect their longitudinal position and re-orient in response to real and simulated displacement along the east-west axis [8, 9]. We have also collected evidence suggesting that reed warblers do not use a time difference mechanism to do so [10]. Instead, we demonstrated that they seem to be able to detect their longitudinal position by measuring magnetic declination, which is the angle between magnetic North and geographic North [4]. This parameter of the geomagnetic field varies along the east-west axis in many parts of the world [4], and taken together with e.g. magnetic inclination or total field strength that many birds are known to be able to perceive [11–15], it forms a bi-coordinate grid that can be used for long-distance navigation on large parts of the Earth's surface.

We recently showed that adult Eurasian reed warblers, i.e. individuals with prior migratory experience, when kept in a declination rotated by 8.5˚ anti-clockwise (change from +5.5˚ to -3˚) during their autumn migration on the eastern Baltic coast and tested in Emlen funnels under clear, moonless, starry skies over the 4–6 days, responded with a dramatic 151˚ change in their mean orientation from WNW to ESE [4]. This change was consistent with re-orientation towards their normal migratory route following the apparent virtual magnetic displacement. Only magnetic declination indicated that the birds had been displaced to a location far (~1450 km) WNW of their normal migratory route. All other environmental cues such as olfactory, celestial and other visual cues indicated that the birds were still on the Baltic coast. In contrast, migratory naïve first-autumn Eurasian reed warblers did not respond by re-orientation to the similar virtual displacement from Rybachy on the eastern Baltic coast to the vicinity of Dundee in Scotland, but instead became disoriented [4].

In this study, we aimed to test how general the effect found in Eurasian reed warblers is. Even though it is often tacitly assumed that orientation and navigation mechanisms are identical in many if not all birds (or at least in night-migratory songbirds), it is not necessarily so [16; compare e.g. 6, 17 and 18, 19]. We therefore tested adult and first-autumn European robins (*Erithacus rubecula*) and adult garden warblers (*Sylvia borin*) in Emlen funnels under the same experimental conditions that led to dramatic reorientation in the adult experienced Eurasian reed warblers [4]. We compared the birds' orientation in a magnetic field with magnetic North rotated 8.5˚ anti-clockwise with the same birds' orientation the natural magnetic field (NMF) conditions in Rybachy on the Baltic coast.

## Material and methods

### Experimental birds and site

All animal procedures (in this case, capture of the birds and simple, non-invasive, behavioral experiments) were approved by the appropriate authorities: Permit 2013–08 by Kaliningrad Regional Agency for Protection, Reproduction and Use of Animal World and Forests; and Permit 2014–12 by the specialized committee of the Scientific Council of the Zoological Institute, Russian Academy of Sciences. We captured adult (n = 17) and juvenile (first-autumn, n = 15) European robins (*Erithacus rubecula*) during autumn migration in September and October 2014 and 2015 at Rybachy (55˚09´ N, 2052´ E), on the Courish Spit, on the

southeastern coast of the Baltic Sea. European robins migrating in autumn through the Courish Spit winter in southwestern Europe [20, 21]. We also captured adult garden warblers (n = 25) at the same location in late summer 2018 and 2019. According to ringing recoveries, garden warblers that migrate through Rybachy in autumn spend their winter in West Africa to which they migrate via southwestern Europe [20–22]. All birds were kept outdoors in individual cages ($40 \times 40 \times 40$ cm) under the local photoperiod. The birds were provided with food (mealworms) and water *ad libitum*. We did not measure the background electromagnetic noise level, which is important for the functionality of the magnetic compass in European robins and garden warblers [23, 24]. It was identical for the control and experimental conditions; furthermore, at the rural testing site on the Courish Spit the anthropogenic noise is likely to have been very low, as suggested e.g. by the functional magnetic compass observed in garden warblers when tested under control conditions at the same site [24, 25].

## Manipulations of the magnetic field

The measured parameters of the natural magnetic field (NMF) of Rybachy were as follows: total intensity 50200 nT ± 80 nT [standard deviation, SD], inclination 70.1˚ ± 0.2˚ [SD], declination +5.5˚ ± 0.1˚ [SD]. The changed magnetic field (CMF) was produced when electric current ran in the same direction through the two subset of windings of the double-wrapped, three-dimensional Merritt four-coil system [26]. The coil system was identical to the ones used by the Oldenburg group and described in detail elsewhere [27]. The same system was used in our earlier experiments on Eurasian reed warblers [4]. CMF differed from NMF in the value of declination only:it was set at -3˚ instead of +5.5˚ (magnetic parameters after virtual magnetic displacement: total intensity 50200 nT ± 100 nT [SD], inclination 70.1˚ ± 0.2˚ [SD], declination -3˚ ± 0.2˚ [SD]). This means that the field was just rotated by 8.5˚ anticlockwise, while total intensity and inclination remained unchanged. The parameters of CMF approximately matched the natural values found in Scotland 20 km west of Dundee (at ~ 56˚30´ N, 3˚20´ W).

## Orientation tests

Orientation tests (3–6 tests per bird) were performed with Emlen funnels [28] outdoors under clear starry skies. All tests were performed in the coil system. During tests in the NMF, the electric current was run antiparallel whereas in the CMF condition electrical current was run in parallel directions within the double-wrapped system. All tests were performed when at least 50% of the starry sky was visible and in most tests, the sky was 95%– 100% clear. Each test lasted 40 min and started at the beginning of astronomical twilight. We used modified Emlen funnels made of aluminium (top diameter 300 mm, bottom diameter 100 mm, slope 45˚ with the top opening covered by netting). The directionality of the birds' activity was recorded as scratches left as birds were hopping in the funnels on a print film covered with a dried mixture of whitewash and glue. Two researchers (AP and AD or FC) independently determined each bird's mean direction from its distribution of scratches. If both observers considered the scratches to be randomly distributed or if the two mean directions deviated by more than 30º, the bird was considered as disoriented in a given test. The mean of the individual bird's directions was recorded as an orientation data point. We included the results of the birds tested at least three and up to five times that showed at least two and up to five results being sufficiently active (i.e. left at least 40 and nearly always >100 scratches on print film) and with orientation sufficiently concentrated according to the Rayleigh test of uniformity [29] at 5% significance level. Inactive birds (<40 scratches) and disoriented individuals (the mean vector not significant) were excluded from analysis. The group mean vectors for each condition were calculated by vector addition of unit vectors in each of the individual birds' mean directions.

## Statistics

The nonparametric Mardia-Watson-Wheeler (MWW) test was used to test for differences in the mean orientation direction between experimental groups. We did not use the more powerful parametric Watson-Williams test, because the r-values for our group mean vectors in many cases were < 0.75. An r value > 0.75 is a crucial assumption for the Watson-Williams test [29]. Results were regarded as significant if $P < 0.05$. Statistical tests were performed with the ORIANA (Kovach Computing Services, version 4.02). To compare the amount of non-active and random trials between the control condition and virtual magnetic displacements, we used Wilcoxon signed-rank test in order to show that virtual displacement did not affect orientation behaviour of experimental birds.

In order to explore the probability of type II error, i.e. that we failed to detect the existing change of direction after virtual magnetic displacement, e.g. due to the low sample size, we performed the following simulation. We rotated all real values obtained in NMF anticlockwise so that the mean value of the new distribution was 102˚, equal to the mean direction shown by Eurasian reed warblers in CMF [4], under the magnetic conditions identical to the ones used in this study. After that, we ran simulations using the bootstrap technique. With this technique, a random sample of n orientation directions is drawn with replacement from the sample of n orientation angles present in the significantly oriented experimental group (n = 17 for the adult European robins, n = 15 for juvenile European robins, n = 25 for adult garden warblers). Based on these orientation angles, we calculated the mean directions of the simulated distributions. This procedure was repeated 100,000 times, each time with a new randomization. In the next step, the resulting 100,000 mean directions were ranked in ascending order. We looked which proportion of these 100,000 values was within the 95% or 99% confidence interval [CI] of the angular distribution of directions obtained in real tests after the virtual magnetic displacement. Additionally, we performed several Monte Carlo simulations. First, to estimate the possible directional shift, we performed multiple comparison of control sample against artificial samples from a theoretical von Mises population. Second, we estimated the 95% CI of mean direction and mean vector size of our samples and evaluated the probability to obtain a sample with the same parameters from a theoretical von Mises population.

## Results

A total of 17 adult European robins were tested for their migratory orientation in Emlen funnels in the NMF and in a CMF rotated 8.5˚ anticlockwise. In the NMF, the birds oriented in the westerly direction ($\alpha = 295˚$ (all directions are indicated relative to geographic north), $r = 0.65$, $n = 17$, $P = 0.001$, 95% CI of a mean group direction [CI_mean] = 269˚–320˚, Fig 1A). The birds that were oriented in CMF showed a similar mean direction ($\alpha = 266˚$, $r = 0.54$, $n = 14$, $P = 0.014$, 95% CI_mean = 229˚–304˚, Fig 1B). The two distributions are statistically indistinguishable (MWW test: $W = 1.19$, $P = 0.55$) and their 95% CI overlap broadly.

We also captured 15 juvenile (first-year) European robins during autumn migration. These birds oriented in the seasonally appropriate direction in the NMF ($\alpha = 242˚$, $r = 0.61$, $n = 15$, $P = 0.003$, 95% CI_mean = 211˚–273˚, Fig 1C) and in the CMF ($\alpha = 219˚$, $r = 0.61$, $n = 14$, $P = 0.003$, Fig 1D (95% CI_mean = 188˚–251˚, Fig 1D). These two distributions are statistically indistinguishable (MWW test: $W = 0.65$, $P = 0.97$) and their 95% CI overlap broadly.

Furthermore, we captured 25 adult garden warblers during autumn migration, of which 20 showed significant orientation in both NMF and CMF. These birds oriented in the seasonally appropriate direction in the NMF ($\alpha = 198˚$, $r = 0.42$, $n = 25$, $P = 0.01$, 95% CI_mean = 162˚–234˚, Fig 1E) and in the CMF ($\alpha = 197˚$, $r = 0.62$, $n = 20$, $P < 0.001$, 95% CI_mean = 172˚–223˚, Fig 1F). These two distributions had nearly identical means and were not significantly

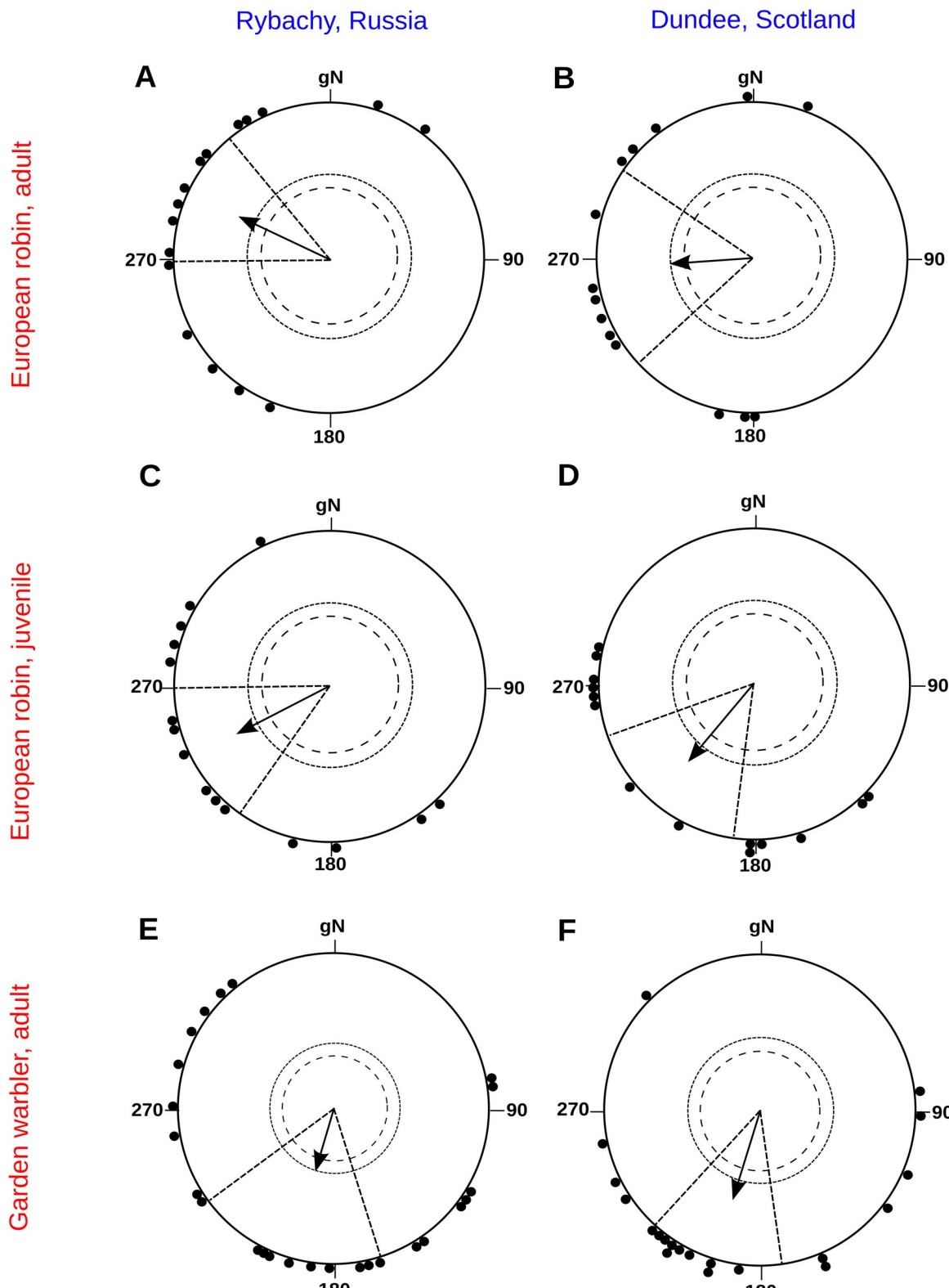

**Fig 1.** Orientation of adult European robins (a, b), juvenile European robins (c, d) and adult garden warblers (e, f) tested during autumn migration on the Courish Spit before (in the natural magnetic field of Rybachy, left column) and after virtual magnetic displacement (in the magnetic field of Scotland, right column). Each triangle at the circle periphery indicates the mean orientation of one individual bird;

arrow shows mean group directions and vector lengths; geographic North corresponds to 0˚. Results of individual tests are presented in S1 Table.

different (MWW test: $W$ = 2.75, $P$ = 0.253). Virtual magnetic displacements did not affect orientation behaviour of either garden warblers or European robins tested in Emlen funnel according to Wilcoxon signed-rank test. The number of non-active and random trials did not differ between control and experimental conditions (Z = 0.77, p = 0.44 and Z = 0.40, p = 0.68, respectively).

If adult European robins were expected to change their direction in CMF and to choose the same mean direction as Eurasian reed warblers did under the same conditions (102˚), all mean values obtained in 100,000 simulations were within the range 29˚–198˚, i.e. outside of the 95% CI_mean and also of the 99% CI_mean (229˚–304˚) of the distribution shown in CMF. For juvenile European robins, the range of simulated mean values was 38˚–156˚, also outside of the 95% CI_mean and of the 99% CI_mean (178˚–261˚) of the distribution shown in CMF. Simulated mean directions of garden warblers of the assumption that they should orient towards 102˚ after the apparent displacement were in the range of 27˚–170˚, which was outside the 95% CI_mean but within the 99% CI_mean (164˚–231˚). Just one value (of 100,000 simulations) was within the 99% CI_mean. The minimum directional shift we could detect with 95% probability using the MWW test was *ca.* 80˚ in garden warblers (S1 Fig) and European robins (S2 Fig) under the assumption that concentration was calculated correctly. Moreover, the probability to obtain samples like ours from different distributions with means at 160˚ and 220˚, respectively, was smaller than 0.16 (S3 and S4 Figs) for adult garden warblers. We did not perform likelihood estimation for European robins due to small sample size.

## Discussion

Our results show no indication of correction (re-orientation towards migratory route [4]) for the apparent virtual westward displacement based on changing the magnetic declination only in either adult (experienced) or first-autumn (naïve) European robins, or in adult garden warblers. These results are in a stark contrast with the results we have earlier obtained in Eurasian reed warblers at the same location, same season and using the same experimental setup [4]. In the latter species, experienced individuals that have migrated down to West Africa (and experienced negative declination values, i.e. when magnetic North is located to the west of the geographic North) at least once, significantly changed their orientation in Emlen funnels in CMF by 151˚ from WSW to ESE, consistent with the explanation that they compensated for the apparent westward displacement. Naïve Eurasian reed warblers responded to the declination change by becoming randomly oriented [4]. Neither European robins nor garden warblers in the similar experiments reported here did anything of this kind.

No experimental group (adult or first-autumn European robins or adult garden warblers) significantly changed their orientation in Emlen funnels in response to an 8.5˚ anticlockwise rotation of the magnetic North. No difference was evident either in the mean tendency or in the scatter, as suggested by the non-significant results of MWW test. Bootstrapping suggested that our results are very robust, because in none of the experimental groups after the apparent magnetic displacement the 95% confidence interval of the observed distribution of directions included any of the 100,000 simulated values expected if the birds corrected for the displacement in a similar manner as the European reed warblers did. In neither group of European robins even the 99% confidence interval included any of the simulated values; and in adult garden warblers just 0.001% of the simulated values were within the 99% confidence interval of the observed distribution.

These results are in contrast to our earlier report that suggested that Eurasian reed warblers respond to the change in magnetic declination alone as if they use declination as a map cue which helps them detecting their position along the East-West axis [4]. In the case of European robins, one might suggest that their navigation system does not involve the use of magnetic declination, because they travel shorter distances than Eurasian reed warblers and winter within Europe and North Africa not crossing the Sahara [30]. This is a rather weak explanation, because in western Europe and northwestern Africa where European robins migrating in autumn through Rybachy spend their winter [20], isolines of magnetic field total intensity and inclination run almost parallel and do not form a grid useful for bi-coordinate navigation [31]. This means that using magnetic declination as a map cue to detect longitude in this part of the world would be a useful option for this species. In any case, our results for garden warblers can certainly not be explained in this manner because the migratory routes and wintering areas of garden warblers and Eurasian reed warblers from the Eastern Baltic are very similar [20, 22]. Assuming that many if not all long-distance, e.g. Palaearctic-African, migrants would benefit from the use of declination, a navigational system useful for a Eurasian reed warbler should also be useful for a garden warbler.

One explanation could be that the navigation systems of most if not all migrants are multi-factorial and make use of all navigation cues available [2]. For some reason, Eurasian reed warblers placed in the CMF but physically remaining on the Baltic coast 'believed' in their apparent westward displacement [4], probably because their long-range navigation system relies on magnetic cues rather heavily and, as suggested by recent data, they do not use e.g. olfactory cues *at this spatial scale* [32]. Relative importance of non-magnetic long-range navigation cues (olfactory or other) might be higher in garden warblers and European robins, so that magnetic displacement on the basis of declination change alone was not sufficient to 'persuade' them that they had been displaced. This diversity of responses seems to parallel the situation with compass systems, where not all species (and populations?) seem to use all potentially available compass cues equally well [16]. It cries for more research into navigational maps of different migrant species, ideally in different parts of the world.

## Supporting information

**S1 Table. Raw data for the results of orientation tests.** This table contains all results of individual orientation tests based on which circular diagrams in Fig 1 were plotted and statistics presented in the Results section were calculated.
(DOC)

**S1 Fig. The probability to make type II error for the garden warbler sample according to Monte-Carlo simulation.** a) The probability to obtain p > 0.2 (according to our result of the MWW test: W = 2.75, P = 0.253) depending on how much the true mean direction of the experimental group (in the CMF) differs from the mean direction of the control group (in the NMF); b) The same for concentrations. Zero value for the mean direction is 197° (the mean direction in the CMF), for the concentration 0.93.
(PNG)

**S2 Fig. The probability to make type II error for adult European robin sample according to Monte-Carlo simulation.** a) The probability to obtain p > 0.5 (according to our result of the MWW test: W = 1.19, P = 0.55) depending on how much the true mean direction of the experimental group (in the CMF) differs from the mean direction of the control group (in the NMF); b) The for concentrations. Zero value for the mean direction is 295° (the mean direction in the CMF), for the concentration 1.74.
(PNG)

**S3 Fig. Estimated likelihood of control garden warbler sample (tested in the NMF) depending on the theoretical distribution this sample belongs to.** This estimation is based on the assumption that the statistical population is from a von Mises population. 95% CI was computed for the mean direction and length of the mean vector of our samples. Colour indicates the probability to obtain a sample with parameters (mean direction and length of the mean vector) included in our confidence intervals, from the von Mises distribution specified on the axes of the plot. Means of theoretical von Mises distribution are shown on X-axis, concentration on Y-axis.
(PNG)

**S4 Fig. Estimated likelihood of our data under the von Mises hypothesis for garden warbler sample in the CMF.** For description and explanation, see the legend to S3 Fig.
(PNG)

## Acknowledgments

The authors are grateful to Anna Anashina who provided technical assistance during the experiments. Sergey Ogurtsov, Vladimir Shakhparonov and Eldar Rakhimberdiev provided helpful advice in statistical data analysis.

## Author Contributions

**Conceptualization:** Alexander Pakhomov, Henrik Mouritsen.

**Data curation:** Alexander Pakhomov.

**Formal analysis:** Nikita Chernetsov, Alexander Pakhomov, Alexander Davydov, Fedor Cellarius.

**Funding acquisition:** Nikita Chernetsov, Alexander Pakhomov, Henrik Mouritsen.

**Investigation:** Alexander Pakhomov, Alexander Davydov, Fedor Cellarius.

**Methodology:** Nikita Chernetsov, Alexander Pakhomov, Henrik Mouritsen.

**Project administration:** Nikita Chernetsov.

**Supervision:** Nikita Chernetsov, Henrik Mouritsen.

**Validation:** Nikita Chernetsov, Alexander Pakhomov.

**Visualization:** Alexander Pakhomov.

**Writing – original draft:** Nikita Chernetsov.

**Writing – review & editing:** Nikita Chernetsov, Alexander Pakhomov, Henrik Mouritsen.

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
