## [Decision Letter · Decision Letter 0]

18 Feb 2020

PONE-D-20-02219

No evidence for the use of magnetic declination for migratory navigation in two songbird species

PLOS ONE

Dear Prof. Chernetsov,

Thank you for submitting your manuscript to PLOS ONE. After careful consideration, we feel that it has merit but does not fully meet PLOS ONE’s publication criteria as it currently stands. Therefore, we invite you to submit a revised version of the manuscript that addresses the points raised during the review process.

We would appreciate receiving your revised manuscript by Apr 03 2020 11:59PM. To enhance the reproducibility of your results, we recommend that if applicable you deposit your laboratory protocols in protocols.io, where a protocol can be assigned its own identifier (DOI) such that it can be cited independently in the future. For instructions see: http://journals.plos.org/plosone/s/submission-guidelines#loc-laboratory-protocols

We look forward to receiving your revised manuscript.

Kind regards,

Ilia Solov'yov

Academic Editor

PLOS ONE

Journal Requirements:

2. In your Methods section, please include a comment about the state of the animals following this research. Were they relased or housed for use in further research?

3. Your ethics statement must appear in the Methods section of your manuscript. If your ethics statement is written in any section besides the Methods, please move it to the Methods section and delete it from any other section. Please also ensure that your ethics statement is included in your manuscript, as the ethics section of your online submission will not be published alongside your manuscript.

Reviewers' comments:

Reviewer's Responses to Questions

**Comments to the Author**

1. Is the manuscript technically sound, and do the data support the conclusions?

Reviewer #1: Yes

Reviewer #2: Yes

2. Has the statistical analysis been performed appropriately and rigorously? 

Reviewer #1: Yes

Reviewer #2: Yes

3. Have the authors made all data underlying the findings in their manuscript fully available?

Reviewer #1: Yes

Reviewer #2: Yes

4. Is the manuscript presented in an intelligible fashion and written in standard English?

Reviewer #1: Yes

Reviewer #2: Yes

5. Review Comments to the Author

Reviewer #1: Review on the manuscript “No evidence for the use of magnetic declination for migratory navigation in two songbird species”.

The sensory mechanisms behind avian map sense remain an exciting and largely unexplored topic. Therefore, every bit of information that we obtain from very time-consuming behavioural experiments is valuable.

In their previous paper, Chernetsov et al. (2017) have shown that Eurasian reed warblers can use magnetic declination to detect their longitudinal position. In their current study the authors aimed to replicate the same virtual magnetic displacement in two other migratory species, long-distance migratory garden warblers and short-distance migratory European robins. Surprisingly, both species did not seem to respond to the declination shift. This result appears contradictory to the previous data on reed warblers, which does not make it less interesting.

I, therefore, strongly recommend this paper to be published. I have, nevertheless, a few minor comments regarding the data analysis and discussion of the results.

1. The major drawback of this study, and I believe, the authors are quite aware of it, is the fact that we do not know the reaction of robins and garden warblers to geographical east-west displacement. As far as I know, apart from Rabøl’s old experiments (with rather contradictory results), we do not know anything about the map sense of these two species.

I expect that, due to the current geopolitical situation, it would be rather problematic to geographically displace birds from Russia to Scotland. However, I disagree with the authors’ decision to assume that the expected direction of garden warblers and robins after the declination shift should be the same as it was in reed warblers (average 102°), since their initial orientation in NMF is already slightly different. I would rather assume that their re-orientation response might be similar, so 151° counter-clockwise. By the way, how exactly did you rotate the values for the modelled distribution: by taking all the single values from all trials, or just individual mean orientation responses?

Moreover, doing the bootstrap comparison between two distributions that would obviously never overlap (they are basically opposite directions) seems like overkill to me. I would rather suggest you to calculate the minimum directional shift that you would be able to detect based on the dispersion of your data.

2. Just to show that the declination shift, indeed, had no influence on birds orientation behaviour, I would suggest to compare the amount of not-active and random trials, general activity (if possible) and directedness (distribution of r-values).

3. Could you, please, include p-level thresholds directly into circular diagrams?

4. Could you provide the standard deviation of the magnetic field parameters during your tests?

Reviewer #2: The study by Chernetsov et al. draws on the previous work of the same lab where it was shown that adult Eurasian reed warblers can use magnetic declination to solve the longitude problem (Chernetsov et al. 2017 Curr Biol) – a navigational issue many human navigators used to struggle with for centuries. In the present manuscript, the authors examined 2 more songbird migrant species (European robins and Garden Warblers) for their use of a declination-based magnetic navigation mechanism. Surprisingly, they did not find any reaction of adult or first-autumn robins or adult garden warblers to the change of declination which led to a dramatic compensatory response (re-orientation towards the migratory route) in adult reed warblers. The methodology and results appear to be very robust as additional bootstrap analysis did not reveal any difference between the orientation results in Natural and Changed magnetic field conditions (NMF vs CMF treatments). The authors point out that these new results suggest that there is a substantial variation in the use of geomagnetic cues for navigation across songbird species which used to be unknown and resembles the recently described variation of bird compass mechanisms (see Chernetsov 2015 for a review). In my opinion, these new findings are very important to humble researchers who tend to generalize small sample findings onto larger set of cases and species. Additionally, these results urge for further investigations of navigational mechanisms in different avian species from different continents to better understand what ecological and evolutionary patterns emerge in this variation of navigation mechanisms. Given that the present study and its results are rigorous and novel, I endorse this manuscript for publication in PLoS One as long as the authors address some minor suggestions listed below:

Abstract

L. 23 the East-West position

Introduction

L. 42 Check British English spelling for consistency throughout. Here ‘kilometres’

L. 46-50. The wording should be improved to something:

“Through geographic latitude…., the measurement of longitude is less straightforward.”

L. 50 ‘did not KNOW…’

L. 82 Please add 2 more highly relevant references to showcase the variation of compass and navigation strategies in birds revealed recently

[16; compare 6

[Thorup et al. 2007 – compensation by displaced Zonothrichia sparrows], and

17 [Cochran et al. 2004 – sun compass calibrates magnetic compass]

Vs.

18 [Chernetsov et al. 2011 – the lack of calibration of magnetic compass by sun one],

19 [add this new reference] Kishkinev et al. 2016 D. Kishkinev, D. Heyers, B.K. Woodworth, G.W. Mitchell, A.H. Keith, D.R. Norris (2016) Experienced migratory songbirds do not display goal-ward orientation after release following a cross-continental displacement: an automated telemetry study Scientific Reports 6, 37326 - the lack of compensation in displaced Zonotrichia sparrows.

L. 84 led to A dramatic re-orientation

L. 86 A magnetic field

Methods

L. 92 – bring the total figures of captured birds of the species/ages used

L. 129 Did you mean AD (Alexander Davydov?), not AA (there are no authors with such initials)

L.134 Disagreement between here and L. 119 (3-6 tests per bird). Re-word

L.137 ‘the Rayleigh test of uniformity’ to be more explicit for non-specialists in circular stats

Results

LL.166-68 Belongs to Methods

Discussion

L. 209. A more specific wording would remind the reader of expectations: ‘Our results show no indication of correction (re-orientation towards migratory route [Chernetsov et al. 2017])…’

L. 212 In A stark contrast

L. 212 Obtained FROM…

L. 234 that adult ERWs…

L. 244 would be a useful option for the species

LL.247-248: The following wording would better elaborate the rationale for this comparison:

‘Assuming that many if not all long-distance, e.g. Palaearctic-African, migrants would benefit from the use of declination, a navigational system useful for a ERW should be also useful for a garden warbler’

L.254: delete the citation in ()

References

L. 278. Year should be 2018

Figure 1 should be improved by doing the following:

- Adding ‘NMF’ and ‘CMF’ and Subheadings ‘European robins (adults)’, ‘ER (first-autumn)’, ‘Garden warblers (adults)’ for 1-3 rows

- Each circular diagram needs to show (i) 1% and 5% significance levels as dashed line circles and (ii) 95% CIs flanking mean group directions

- For better quality of the figure, I would strongly recommend to use vector graphic editor (e.g. by using Inkscape or other popular vector graphic editor + rasterizing export) and then high-resolution rasterizing rather than just default raster low-resolution Oriana pictures. The latter could be used as templates for the former.

6. PLOS authors have the option to publish the peer review history of their article (what does this mean?). If published, this will include your full peer review and any attached files.

Reviewer #1: No

Reviewer #2: No

---

## [Author Response · Author response to Decision Letter 0]

30 Mar 2020

Reviewer #1: Review on the manuscript “No evidence for the use of magnetic declination for migratory navigation in two songbird species”.

The sensory mechanisms behind avian map sense remain an exciting and largely unexplored topic. Therefore, every bit of information that we obtain from very time-consuming behavioural experiments is valuable.

In their previous paper, Chernetsov et al. (2017) have shown that Eurasian reed warblers can use magnetic declination to detect their longitudinal position. In their current study the authors aimed to replicate the same virtual magnetic displacement in two other migratory species, long-distance migratory garden warblers and short-distance migratory European robins. Surprisingly, both species did not seem to respond to the declination shift. This result appears contradictory to the previous data on reed warblers, which does not make it less interesting.

I, therefore, strongly recommend this paper to be published. I have, nevertheless, a few minor comments regarding the data analysis and discussion of the results.

1. The major drawback of this study, and I believe, the authors are quite aware of it, is the fact that we do not know the reaction of robins and garden warblers to geographical east-west displacement. As far as I know, apart from Rabøl’s old experiments (with rather contradictory results), we do not know anything about the map sense of these two species.

I expect that, due to the current geopolitical situation, it would be rather problematic to geographically displace birds from Russia to Scotland. However, I disagree with the authors’ decision to assume that the expected direction of garden warblers and robins after the declination shift should be the same as it was in reed warblers (average 102°), since their initial orientation in NMF is already slightly different. I would rather assume that their re-orientation response might be similar, so 151° counter-clockwise. By the way, how exactly did you rotate the values for the modelled distribution: by taking all the single values from all trials, or just individual mean orientation responses?

Moreover, doing the bootstrap comparison between two distributions that would obviously never overlap (they are basically opposite directions) seems like overkill to me. I would rather suggest you to calculate the minimum directional shift that you would be able to detect based on the dispersion of your data.

We completely agree with the reviewer that we do not know how European robins and garden warblers respond to both physical and virtual magnetic displacements because these species were not used in such experiments before, in contrast to Eurasian reed warblers. Currently, we know only that reed warblers show re-orientation after virtual magnetic displacements and it has been shown in at least two populations of this species during spring and autumn experiments in Russia (Kishkinev et al., 2015; Chernetsov et al., 2017; Pakhomov et al., 2018) and autumn experiments in Austria (Packmor et al. 2019: Evidence for true magnetic navigation in a long-distance migratory songbird. Abstracts of 12th European Ornithologists’ Union, Cluj-Napoca, Romania, p. 111-112). 

Our prediction was that the response of these species to such displacement would be similar to the results of experiments in Eurasian reed warblers (Chernetsov et al., 2017). Therefore, when we performed bootstrap analysis, we created the modelled distribution by taking individual means and rotating them 151° counter-clockwise. Additionally to previous bootstrap analysis mentioned in the reviewed version of the manuscript, we performed several estimations using Monte Carlo machinery: type II error probability according to the MWW test depends on effect size; likelihood of our data assuming the hypothesis of von Mises population and added that in the manuscript in the following sections:

Methods:

Lines 168-173

Additionally, we performed several Monte Carlo simulations. First, to estimate the possible directional shift, we performed multiple comparison of control sample against artificial samples from a theoretical von Mises population. Second, we estimated the 95% CI of mean direction and mean vector size of our samples and evaluated the probability to obtain a sample with the same parameters from a theoretical von Mises population.

Results:

Lines 219-224

The minimum directional shift we could detect with 95% probability using the MWW test was ca. 80° in garden warblers (S1 Fig.) and European robins (S2 Fig.) under the assumption that concentration was calculated correctly. Moreover, the probability to obtain samples like ours from different distributions with means at 160° and 220°, respectively, was smaller than 0.16 (S3 Fig. and S4 Fig.) for adult garden warblers. We did not perform likelihood estimation for European robins due to small sample size.

Supplementary: 

S1 Fig., S2 Fig., S3 Fig., S4 Fig.

2. Just to show that the declination shift, indeed, had no influence on birds orientation behaviour, I would suggest to compare the amount of not-active and random trials, general activity (if possible) and directedness (distribution of r-values).

We compared the amount of non-active and random trials in NMF and CMF using Wilcoxon signed-rank test and added results of this comparison in the main text of manuscript:

Methods:

Lines 149-152: To compare the amount of non-active and random trials between the control condition and virtual magnetic displacements, we used Wilcoxon signed-rank test in order to show that virtual displacement did not affect orientation behaviour of experimental birds. 

Results: 

Lines 204-208:

Virtual magnetic displacements did not affect orientation behaviour of either garden warblers or European robins tested in Emlen funnel according to Wilcoxon signed-rank test. The number of non-active and random trials did not differ between control and experimental conditions (Z = 0.77, p = 0.44 and Z = 0.40, p = 0.68, respectively).

Unfortunately, we cannot compare general activity of birds in the NMF and the CMF conditions because we did not save raw data of each test (the amount of scratches which bird left on a print film in each orientation test). 

3. Could you, please, include p-level thresholds directly into circular diagrams?

Done

4. Could you provide the standard deviation of the magnetic field parameters during your tests?

Done

We added the calculation of SD in the manuscript:

Lines 166-188:

The measured parameters of the natural magnetic field (NMF) of Rybachy were as follows: total intensity 50200 nT ± 80 nT [standard deviation, SD], inclination 70.1° ± 0.2° [SD], declination +5.5° ± 0.1° [SD]…. CMF differed from NMF in the value of declination only: it was set at -3° instead of +5.5° (magnetic parameters after virtual magnetic displacement: total intensity 50200 nT ± 100 nT [SD], inclination 70.1° ± 0.2° [SD], declination - 3° ± 0.2° [SD]).

Reviewer #2: The study by Chernetsov et al. draws on the previous work of the same lab where it was shown that adult Eurasian reed warblers can use magnetic declination to solve the longitude problem (Chernetsov et al. 2017 Curr Biol) – a navigational issue many human navigators used to struggle with for centuries. In the present manuscript, the authors examined 2 more songbird migrant species (European robins and Garden Warblers) for their use of a declination-based magnetic navigation mechanism. Surprisingly, they did not find any reaction of adult or first-autumn robins or adult garden warblers to the change of declination which led to a dramatic compensatory response (re-orientation towards the migratory route) in adult reed warblers. The methodology and results appear to be very robust as additional bootstrap analysis did not reveal any difference between the orientation results in Natural and Changed magnetic field conditions (NMF vs CMF treatments). The authors point out that these new results suggest that there is a substantial variation in the use of geomagnetic cues for navigation across songbird species which used to be unknown and resembles the recently described variation of bird compass mechanisms (see Chernetsov 2015 for a review). In my opinion, these new findings are very important to humble researchers who tend to generalize small sample findings onto larger set of cases and species. Additionally, these results urge for further investigations of navigational mechanisms in different avian species from different continents to better understand what ecological and evolutionary patterns emerge in this variation of navigation mechanisms. Given that the present study and its results are rigorous and novel, I endorse this manuscript for publication in PLoS One as long as the authors address some minor suggestions listed below:

Abstract

L. 23 the East-West position

Done

Introduction

L. 42 Check British English spelling for consistency throughout. Here ‘kilometres’

Done

L. 46-50. The wording should be improved to something:

“Through geographic latitude…., the measurement of longitude is less straightforward.”

Done

Lines 47-50

Through geographic latitude can be inferred e.g. from the elevation of the centre of stellar rotation (e.g. the North Star) on any date, the elevation of the sun at solar noon (when considering season), or from magnetic inclination, geographic longitude is less straightforward to measure

L. 50 ‘did not KNOW…’

Done

L. 82 Please add 2 more highly relevant references to showcase the variation of compass and navigation strategies in birds revealed recently

[16; compare 6

[Thorup et al. 2007 – compensation by displaced Zonothrichia sparrows], and

17 [Cochran et al. 2004 – sun compass calibrates magnetic compass]

Vs.

18 [Chernetsov et al. 2011 – the lack of calibration of magnetic compass by sun one],

19 [add this new reference] Kishkinev et al. 2016 D. Kishkinev, D. Heyers, B.K. Woodworth, G.W. Mitchell, A.H. Keith, D.R. Norris (2016) Experienced migratory songbirds do not display goal-ward orientation after release following a cross-continental displacement: an automated telemetry study Scientific Reports 6, 37326 - the lack of compensation in displaced Zonotrichia sparrows.

Done

L. 84 led to A dramatic re-orientation

Done

L. 86 A magnetic field

Done

Methods

L. 92 – bring the total figures of captured birds of the species/ages used

Done

L. 129 Did you mean AD (Alexander Davydov?), not AA (there are no authors with such initials)

Done. It was Alexander Davydov. 

Two researchers (AP and AD or FT) independently determined each bird’s mean direction from its distribution of scratches.

L.134 Disagreement between here and L. 119 (3-6 tests per bird). Re-word

Done. 

Orientation tests (3–5 tests per bird) were performed with Emlen funnels (Emlen and Emlen, 1966) outdoors under clear starry skies.

L.137 ‘the Rayleigh test of uniformity’ to be more explicit for non-specialists in circular stats

Done.

We included the results of the birds tested at least three and up to five times that showed at least two and up to five results being sufficiently active (i.e. left at least 40 and nearly always >100 scratches on print film) and with orientation sufficiently concentrated according to the Rayleigh test of uniformity (Batschelet, 1981) at 5% significance level.

Results

LL.166-68 Belongs to Methods

A total of 17 adult European robins were tested for their migratory orientation in Emlen funnels in the NMF and in a CMF rotated 8.5° anticlockwise

Discussion

L. 209. A more specific wording would remind the reader of expectations: ‘Our results show no indication of correction (re-orientation towards migratory route [Chernetsov et al. 2017])…’

Done

L. 212 In A stark contrast

Done

L. 212 Obtained FROM…

We think that the use of IN is more preferable than FROM in this case.

L. 234 that adult ERWs…

We decided to use full name of species (Eurasian reed warbler) instead of short name (ERWs) as a part of common style of our manuscript. 

L. 244 would be a useful option for the species

Done

LL.247-248: The following wording would better elaborate the rationale for this comparison:

‘Assuming that many if not all long-distance, e.g. Palaearctic-African, migrants would benefit from the use of declination, a navigational system useful for a ERW should be also useful for a garden warbler’

Done

L.254: delete the citation in ()

Done, thank you for spotting this.

References

L. 278. Year should be 2018

Done (but in Line 280)

2. Mouritsen H. Long-distance navigation and magnetoreception in migratory animals. Nature. 2018; 55: 50–59. doi: 10.1038/s41586-018-0176-1

Figure 1 should be improved by doing the following:

- Adding ‘NMF’ and ‘CMF’ and Subheadings ‘European robins (adults)’, ‘ER (first-autumn)’, ‘Garden warblers (adults)’ for 1-3 rows

- Each circular diagram needs to show (i) 1% and 5% significance levels as dashed line circles and (ii) 95% CIs flanking mean group directions

- For better quality of the figure, I would strongly recommend to use vector graphic editor (e.g. by using Inkscape or other popular vector graphic editor + rasterizing export) and then high-resolution rasterizing rather than just default raster low-resolution Oriana pictures. The latter could be used as templates for the former.

We redrew our figure in the manuscript according to your comments and suggestions.

---

## [Decision Letter · Decision Letter 1]

8 Apr 2020

No evidence for the use of magnetic declination for migratory navigation in two songbird species

PONE-D-20-02219R1

Dear Dr. Chernetsov,

We are pleased to inform you that your manuscript has been judged scientifically suitable for publication and will be formally accepted for publication once it complies with all outstanding technical requirements.

With kind regards,

Ilia Solov'yov

Academic Editor

PLOS ONE

Additional Editor Comments (optional):

Reviewers' comments:

Reviewer's Responses to Questions

**Comments to the Author**

1. If the authors have adequately addressed your comments raised in a previous round of review and you feel that this manuscript is now acceptable for publication, you may indicate that here to bypass the “Comments to the Author” section, enter your conflict of interest statement in the “Confidential to Editor” section, and submit your "Accept" recommendation.

Reviewer #1: All comments have been addressed

Reviewer #2: All comments have been addressed

2. Is the manuscript technically sound, and do the data support the conclusions?

Reviewer #1: Yes

Reviewer #2: Yes

3. Has the statistical analysis been performed appropriately and rigorously? 

Reviewer #1: Yes

Reviewer #2: Yes

4. Have the authors made all data underlying the findings in their manuscript fully available?

Reviewer #1: Yes

Reviewer #2: Yes

5. Is the manuscript presented in an intelligible fashion and written in standard English?

Reviewer #1: Yes

Reviewer #2: Yes

6. Review Comments to the Author

Reviewer #1: I am very pleased that authors have made such an effort to improve the statistical analysis of the data. I do believe that the new analysis is quite important not only for this study, but also for future studies with the same type of data. So I would ask authors to go into a bit more detail on how they performed the analysis in the final version of the manuscript: e.g. name the program used (Matlab?) and, perhaps, provide the script in the supplementary material.

Reviewer #2: The authors have done a brilliant work. They fully addressed all my questions and, as I cross-checked, they addressed the issues raised by Reviewer #1 (and I fully agree with the latter ones). The quality of the revised MS has been significantly improved as well. With that I have no further questions and endorse this revision for acceptance.

7. PLOS authors have the option to publish the peer review history of their article (what does this mean?). If published, this will include your full peer review and any attached files.

Reviewer #1: Yes: Dmitry Kobylkov

Reviewer #2: No

---

## [Editor Report · Acceptance letter]

13 Apr 2020

PONE-D-20-02219R1 

No evidence for the use of magnetic declination for migratory navigation in two songbird species 

Dear Dr. Chernetsov:

I am pleased to inform you that your manuscript has been deemed suitable for publication in PLOS ONE. Congratulations! Your manuscript is now with our production department. 

With kind regards,

on behalf of

Prof. Ilia Solov'yov 

Academic Editor

PLOS ONE